# Sex-Specific Differences in Cytokine Induction by the Glycolipid Adjuvant 7DW8-5 in Mice

**DOI:** 10.3390/biom13010008

**Published:** 2022-12-21

**Authors:** Felicia N. Watson, Caroline J. Duncombe, Anya C. Kalata, Ethan Conrad, Sumana Chakravarty, B. Kim Lee Sim, Stephen L. Hoffman, Moriya Tsuji, Melanie J. Shears, Sean C. Murphy

**Affiliations:** 1Graduate Program in Pathobiology, Department of Global Health, University of Washington, Seattle, WA 98109, USA; 2Department of Laboratory Medicine and Pathology, University of Washington, Seattle, WA 98109, USA; 3Center for Emerging and Re-Emerging Infectious Diseases (CERID), University of Washington, Seattle, WA 98109, USA; 4Sanaria Inc., 9800 Medical Center Drive, Suite A209, Rockville, MD 20850, USA; 5Aaron Diamond AIDS Research Center, Division of Infectious Diseases, Department of Medicine, Columbia University Irving Medical Center, New York, NY 10032, USA; 6Department of Microbiology, University of Washington, Seattle, WA 98109, USA; 7Washington National Primate Research Center, University of Washington, Seattle, WA 98109, USA; 8Department of Laboratories, Seattle Children’s Hospital, Seattle, WA 98109, USA

**Keywords:** 7DW8-5, adjuvant, sex, *Plasmodium*, malaria, vaccine, sporozoites, intradermal

## Abstract

7DW8-5 is a potent glycolipid adjuvant that improves malaria vaccine efficacy in mice by inducing IFN-γ and increasing protective CD8^+^ T cell responses. The addition of 7DW8-5 was previously shown to improve the efficacy of a CD8^+^ T cell-mediated heterologous ‘prime-and-trap’ malaria vaccine against *Plasmodium yoelii* sporozoite challenge in inbred female mice. Here, we report significant differential sex-specific responses to 7DW8-5 in inbred and outbred mice. Male mice express significantly less IFN-γ and IL-4 compared to females following intravenous 7DW8-5 administration. Additionally, unlike in female mice, 7DW8-5 did not improve the vaccine efficacy against sporozoite challenge in prime-and-trap vaccinated male mice. Our findings highlight the importance of including both female and male sexes in experimental adjuvant studies.

## 1. Introduction

Adjuvants are compounds that are added to vaccines to enhance immune responses. Currently very few adjuvants have been licensed for clinical vaccination and the majority of these have a bias toward enhancing humoral and helper T cell immune responses (reviewed in [1]). There is a critical lack of approved adjuvants that enhance cytotoxic CD8^+^ T cell responses to vaccination [1]. Importantly, many pathogens, including the *Plasmodium* parasites that cause malaria, are best controlled by the host through mechanisms that include cytotoxic CD8^+^ cell responses [2]. Thus, more adjuvants, and especially those that improve cytotoxic CD8^+^ T cell responses, will likely be needed to improve the efficacy of vaccines targeting these complex pathogens.

Glycolipid adjuvants that stimulate *i*NKT cells are of increasing interest due to their ability to activate not only humoral immunity, but also cytotoxic CD8^+^ T cells (reviewed in [3]). α-Galactosylceramide (α-GalCer) is a glycolipid adjuvant that binds CD1d on antigen presenting cells (APCs), activates *i*NKT cells, and induces a cascade of immune cell activation that improves vaccine-induced immune responses [4,5,6]. α-GalCer has been studied for many years and clinical trials with this compound demonstrated safety and potent enhancement of immune responses in humans (reviewed in [7]). However, several clinical trials with α-GalCer have shown suboptimal results, which inspired the development of synthetic α-GalCer analogs. There are now hundreds of α-GalCer analogs that have been developed, which have different toxicological profiles and/or immunostimulatory effects [8]. 

7DW8-5 is a synthetic analog of α-GalCer that has significant translational potential. 7DW8-5 has been tested as an adjuvant and therapy for a variety of infectious diseases and cancer (reviewed in [9]). 7DW8-5 binds CD1d with higher affinity than α-GalCer and has shown higher efficacy at a 100-fold lower dose compared to α-GalCer in rodent and in vitro human cell models [10]. Importantly, 7DW8-5 increases the efficacy of live-attenuated malaria sporozoite (spz) vaccines by increasing CD8^+^ T cell responses [11,12]. However, the majority of pre-clinical rodent studies involving 7DW8-5 were completed in female inbred mice. Since substantial sex-specific differences in immune responses to adjuvants including α-GalCer have been noted [13,14], we sought to directly compare the impact of biological sex on 7DW8-5 adjuvant effects.

## 2. Materials & Methods

*Mice*: Male and female inbred BALB/cJ or Swiss outbred J:ARC(S) mice were purchased at 4–6 weeks of age from Jackson Laboratories (Barr Harbor, ME, USA). All animals were housed at the University of Washington Institutional Animal Care and Use Committee (IACUC)-approved animal facility and were used under an approved IACUC protocol (4317-01 to S.C.M.).

*7DW8-5:* 7DW8-5 was gifted from M. Tsuji and prepared as described [12]. All mice received 2 μg 7DW8-5 administered via intravenous (IV) or intradermal (ID) route. IV injections were administered retro-orbitally in 100 µL with an Exel International Insulin Syringes with a 29 G permanently attached needle (product #26018). ID injections were administered on the lower back near the base of the tail in two 10 µL injections with a BD Veo Insulin Syringe with Ultra-Fine needle 6 mm × 31 G 3/10 mL/cc (product #324909). 

*ELISA:* Plasma was isolated from mice and IFN-γ and IL-4 cytokines were measured with commercially available ELISA kits (BioLegend, San Diego, CA, USA) as previously described [12]. All plasma was frozen at −80 °C in single-use aliquots.

*Plasmodium yoelii DNA prime-and-RAS trap immunization and challenge:* Prime-and-trap vaccines were prepared as previously described with minor modifications described below [12]. Briefly, a DNA insert encoding the full-length *Plasmodium yoelii* (Py) circumsporozoite protein (CSP) without the major repeat region (248aa: MKKCTILVVASLLLVDSLLPGYGQNKSVQAQRNLNELCYNEENDNKLYHVLNSKNGKIYNRNIVNRLLGDALNGKPEEKKDDPPKDGNKDDLPKEEKKDDLPKEEKKDDPPKDPKKDDPPKEAQNKLNQPVVADENVDQ|PRPQPDGNNNNNNNNGNNNEDSYVPSAEQILEFVKQISSQLTEEWSQCSVTCGSGVRVRKRKNVNKQPENLTLEDIDTEICKMDKCSSIFNIVSNSLGFVILLVLVFFN) was cloned into the pUb.3 vector, loaded onto gold beads, and administered to mice via gene gun (ggCSP). One month later, 2 × 10^4^ total cryopreserved Py radiation attenuated sporozoites (cryo-RAS, Sanaria Inc., Rockville, MD, USA) with or without 7DW8-5 were administered ID into the rear left footpad in two 2.5 µL injections with a NanoFil syringe (World Precision Instruments, LLC, Sarasota, FL, USA). Cryo-RAS and 7DW8-5 were mixed immediately before injection. Animals were challenged IV via the retro-orbital route with 1 × 10^3^ freshly dissected wild-type infectious Py spz (obtained from Seattle Children’s Research Institute, Seattle, WA, USA) and monitored for the presence of blood stage parasites by Giemsa stain thin blood smear microscopy for 14 days.

*Statistics:* ELISA data was analyzed with non-parametric Mann–Whitney tests. All error bars represent the standard deviation (SD) of the mean with individual mouse samples shown, if applicable. Protection data was analyzed with Fisher Exact test. *p* values above 0.05 were considered not significant (ns). GraphPad Prism 9.1.2 Software (San Diego, CA, USA) was used for all calculations.

## 3. Results

As a first step towards evaluating the impact of biological sex on 7DW8-5 adjuvant effects, we reviewed prior literature on 7DW8-5 and searched for information on research subject sex. A total of 17 papers were identified by searching “7DW8-5” on PubMed^®^. Two papers were excluded -- one contained no animal experiments [15] and one was a review article with limited primary research data [16]. Of the 15 remaining papers, 13 used 7DW8-5 in rodent models, one used 7DW8-5 in non-human primates (NHP), and one used 7DW8-5 with human ex vivo samples (Table 1). The single study in NHPs used only male animals, while only 9 of the 13 papers in rodent models listed the sex of the animals, and all of these used female mice (Table 1). We also noted that these studies of 7DW8-5 in rodent models were also exclusively performed in inbred mouse models. Thus, we concluded there was a need for further evaluation of the impact of biological sex on 7DW8-5 adjuvant effects in pre-clinical research models. 

We previously reported that IV administration of 2 μg 7DW8-5 induced potent transient levels of IFN-γ cytokines in female inbred BALB/cJ mouse blood with strong responses peaking at 12 h and declining by 24 h [12]. To assess the impact of biological sex on this phenomenon, we repeated the same experiment in male BALB/cJ mice and found an ~6-fold reduction in IFN-γ induced by IV 7DW8-5 administration in male mice compared to female mice at the 12 h timepoint (Figure 1). 7DW8-5 is known to preferentially induce Th1 cytokines (e.g., IFN-γ), but we have also shown that detectable levels of Th2 cytokines (e.g., IL-4) are also induced by IV 7DW8-5 administration with similar expression kinetics [12]. Since 7DW8-5 is also known to induce IL-4 production, we also assessed IL-4 levels in the same mice. We found that there was no detectable IL-4 induced by IV 7DW8-5 administration in male mice, but we did detect IL-4 in female mice at the 12 h timepoint (Figure 1). Thus, male and female BALB/cJ mice show striking sex-specific differences in the levels of these plasma cytokines in response to IV 7DW8-5 administration.

Non-systemic administration routes are also of interest. This is because non-IV administration of 7DW8-5 induced little to no systemic cytokine responses in female mice [11], which may be expected to translate to a better safety profile for clinical use. Therefore, we next evaluated the impact of sex on ID 7DW8-5 administration in mice. We administered 2 µg 7DW8-5 ID to male and female BALB/cJ mice and assessed IFN-γ and IL-4 by ELISA as above. As expected, the ID route induced significantly lower levels of these cytokines, but female mice still had higher levels than males at the 12 h timepoint (Figure 1). Thus, male and female BALB/cJ mice also show sex-specific differences in the levels of these plasma cytokines in response to 7DW8-5 non-systemic ID administration.

As it was possible that these findings were a phenomenon of the BALB/cJ mouse model, we further wanted to investigate if this sex-specific phenotype was apparent in another mouse model. To assess whether these findings were generalizable to outbred mouse strains, we next compared IFN-γ induction following systemic IV or local ID administration of 7DW8-5 in Swiss outbred mice. As seen in the inbred BALB/cJ mice, we found higher levels of this cytokine in females compared to male mice following IV 7DW8-5 administration at 6 h post-injection (Figure 2). As expected, only low levels of IFN-γ were induced by ID 7DW8-5 administration in these mice. Thus, male and female genetically diverse Swiss outbred mice also showed sex-specific differences in the levels of plasma IFN-γ in response to 7DW8-5. 

Lastly, we wanted to determine if 7DW8-5 would be an effective adjuvant in male BALB/cJ mice. Previous literature has demonstrated that live RAS malaria vaccines are improved by the co-administration of 7DW8-5 in female mice [11]. Additionally, our group also demonstrated that 7DW8-5 improved a heterologous two step DNA-prime-and-RAS trap malaria vaccine strategy in female BALB/cJ mice [12]. Moreover, recent data from our group found that 7DW8-5 co-administration with ID-RAS was required to achieve 100% protection in female BALB/cJ mice (In preparation and [29]). Here, we evaluated the efficacy of the same DNA-prime and 7DW8-5-adjuvanted ID-RAS trap vaccine in male BALB/cJ mice. Strikingly, we found that 7DW8-5 did not significantly improve vaccine efficacy, with poor protection seen following DNA-prime and ID-RAS trap groups both with and without 7DW8-5 (Figure 3). Thus, the adjuvant effects of 7DW8-5 are drastically different in male and female mice when used in this malaria vaccine strategy.

## 4. Discussion

Novel adjuvants are critically needed for the clinic. Glycolipids are potent adjuvants that have been shown to have potent immunostimulatory effects, including the ability to induce cytotoxic CD8^+^ T cell responses [3]. As such, they are of increasing interest for use in vaccines targeting complex pathogens like *Plasmodium*. 7DW8-5 is of particular interest due its ability to activate *i*NKT cells and induce Th1-like responses, including the activation of CD8^+^ T cells, and to do so at lower doses than α-GalCer [10]. Previous reports in rodent models only examined adjuvant effects of 7DW8-5 in female inbred mice. However, since significant biological sex-based differences have been noted for α-GalCer [13,14], here we sought to investigate if 7DW8-5 similarly induced different adjuvant effects in male and female mice. We found that IFN-γ was induced by IV 7DW8-5 administration in both inbred and outbred mice, but the cytokine levels were significantly lower in males compared to females. Additionally, we found that unlike in females, 7DW8-5 did not improve the efficacy of a DNA prime-and-RAS trap malaria vaccine in male BALB/cJ mice. Taken together, we demonstrate that 7DW8-5 adjuvant effects can be drastically different in male and female mice.

Our findings are supported by other studies that found significant sex-specific differences in the *i*NKT immune responses of inbred C57BL/6 mice [13,14]. Although both reports concluded that IFN-γ cytokine levels induced by glycolipids were higher in female mice than male mice, these prior studies proposed different mechanisms and contributions from sex hormones. Gourdy et al. suggested that estrogens but not testosterone were responsible for the differential immune responses [13], while Lotter et al. suggested that testosterone moderated cytokine production of *i*NKT cells in mice [14]. Based on this data, we hypothesize that in our model, high testosterone in male mice inhibits the induction of IFN-γ following 7DW8-5 administration. Future efforts in the laboratory will seek to investigate the role of estrogens and androgens in mouse models of prime-and-trap malaria vaccination. A limitation of this study is that we did not account for weight differences between male and female mice. On average, six-week-old BALB/cJ male mice weigh more than age matched female mice (male: 23.7 g SD 1.4 vs. female: 19.1 g SD 1.2) [30]. Thus, female mice are receiving a higher dose (mg/kg) of adjuvant. It is possible that a weight normalized adjuvant dose would minimize the differences in cytokine induction that we observed. We also noted that the experiments in Swiss outbred mice resulted in greater variation of the data, which was not unexpected since outbred mice are more genetically diverse and may better represent how diverse human populations may respond to adjuvants. As a future pre-clinical step, our understanding of 7DW8-5 will also greatly benefit from studies using NHP models of malaria cryo-RAS vaccination and challenge. 

In 2016, the NIH introduced a policy mandating inclusion of sex as a biological variable in all vertebrate animals and human studies [31,32,33]. Our findings here indeed highlight the importance of including both biological sexes in experimental adjuvant studies, documenting sex-specific data in publications, and considering how biological sex may impact pre-clinical and clinical outcomes. Strengthened efforts to investigate sex as a biological variable in pre-clinical research studies may be especially important for translating research findings from rodent models to larger NHP models and ultimately to humans.

## Figures and Tables

**Figure 1 biomolecules-13-00008-f001:**
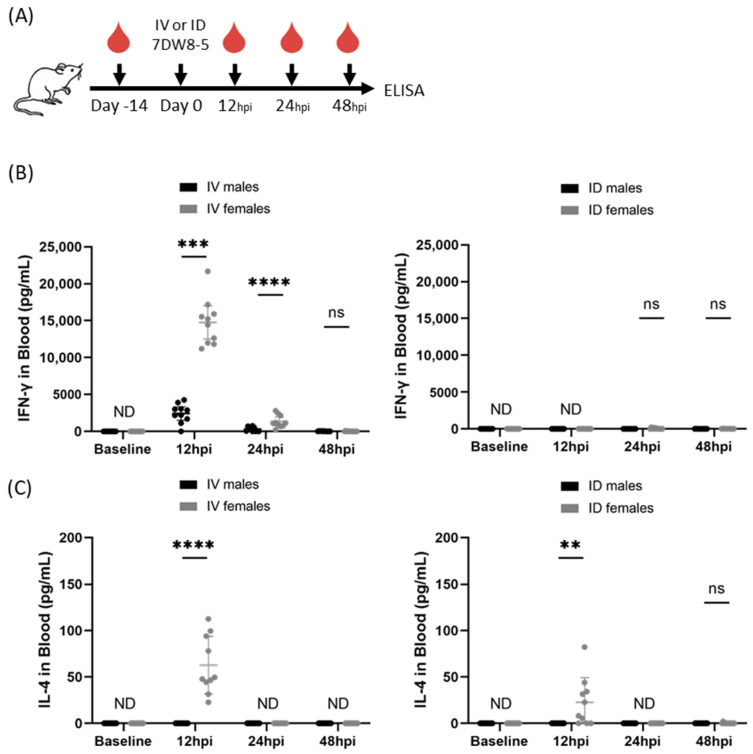
7DW8-5 induces higher levels of systemic IFN-γ and IL-4 in female vs. male BALB/cJ mice. (**A**) Experimental design of blood plasma ELISA studies. IFN-γ (**B**) or IL-4 (**C**) cytokine levels induced by IV-7DW8-5 (**left**) or ID-7DW8-5 (**right**) in mouse blood plasma. Female IV data adapted from Watson et al. for comparison [12]. Error bars represent the SD of the mean of N = 10 BALB/cJ mice across two independent experiments. ELISA data analyzed with Mann–Whitney Tests, **** *p* < 0.0001, *** *p* < 0.001, ** *p* < 0.01, ns *p* > 0.05. ND = Not detected. ID = two 10 μL injections for all.

**Figure 2 biomolecules-13-00008-f002:**
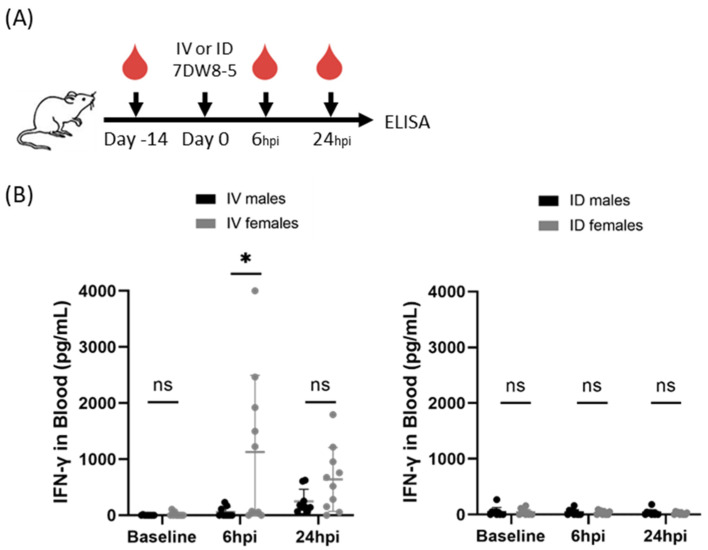
7DW8-5 induces higher levels of systemic IFN-γ in female vs. male Swiss outbred mice. (**A**) Experimental design of blood plasma ELISA studies. (**B**) IFN-γ cytokine levels induced by IV-7DW8-5 (**left**) or ID-7DW8-5 (**right**) in mouse blood plasma. Error bars represent the SD of the mean of N = 9–10 Swiss outbred mice across two independent experiments. ELISA data analyzed with Mann–Whitney Tests, * *p* < 0.05, ns *p* > 0.05. ID = two 10 μL injections for all.

**Figure 3 biomolecules-13-00008-f003:**
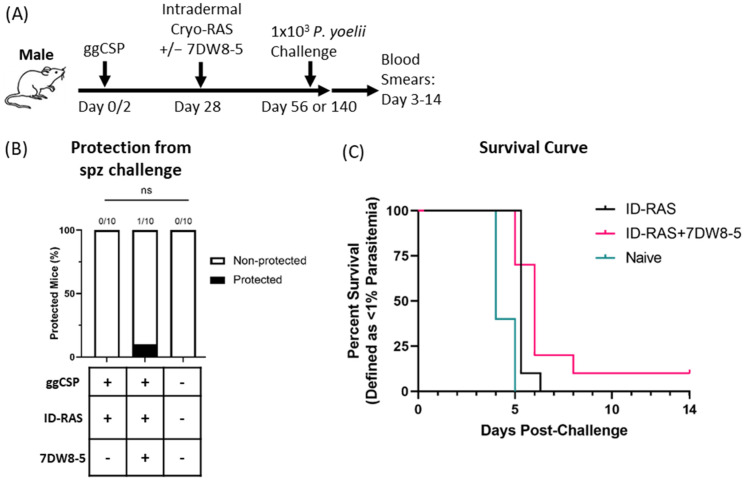
Prime-and-trap vaccination is not protective in male BALB/cJ mice. (**A**) Experimental design of prime-and-trap protection studies. (**B**) Results of protection studies after challenge with 1 × 10^3^ WT purified Py spz administered four weeks after trapping with 2 × 10^4^ ID administered cryo-RAS +/− 7DW8-5. Protection data was from N = 10 male BALB/cJ mice across two independent experiments and analyzed with Fisher Exact Test, ns *p* > 0.05. ID = two 2.5 μL injections for all. (**C**) Survival curve of mice from (**B**).

**Table 1 biomolecules-13-00008-t001:** Research subject sex in published research articles with 7DW8-5.

	Female	Male	Sex Matched	Not Listed	Total	References
Mouse	9	0	0	4	13	[10,11,12,17,18,19,20,21,22,23,24,25,26]
NHP	0	1	0	0	1	[27]
Human	0	0	1	0	1	[28]

## Data Availability

Data sharing not applicable, all data is contained within the article.

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
