# Peer review of "Sex-Specific Differences in Cytokine Induction by the Glycolipid Adjuvant 7DW8-5 in Mice"

_biomolecules, 2022, doi:10.3390/biom13010008_

Round 1

Reviewer 1 Report

Watson et al. reported the importance of inclusion of both female and male sexes in experimental adjuvant studies. The work is concise however clearly pointed out the significance of the experimental design using both sexes. The manuscript is well-written and easy to follow. I have only a few minor comments to improve this manuscript.

Comments:

1) Line 77,

"DNA plasmids" means DNA Vaccine Plasmids, right? Please add brief explanation about the DNA Vaccine plasmid used.

2) Figure 3

How to administer Cryo-RAS and IV 7DW8-5? mixed then iv injected or other way?

Author Response

Line 77, "DNA plasmids" means DNA Vaccine Plasmids, right? Please add brief explanation about the DNA Vaccine plasmid used 

We expanded the methods section about DNA vaccine plasmids to clarify the vaccination methods. 

Figure 3: How to administer Cryo-RAS and IV 7DW8-5? mixed then iv injected or other way? 

We corrected a typo in Figure 3a that may have led to some confusion. The cryo-RAS vaccinations for this study were exclusively administered intradermally (ID). We also expanded the methods section to include more information about the co-administration of cryo-RAS and 7DW8-5 adjuvant. 

Reviewer 2 Report

Watson et al., submitted the study entitled "Sex-specific differences in cytokine induction by the glycolipid adjuvant 7DW8-5 in mice". 

This study highlights the importance of sex differences in efficacy evaluation both in preclinical and clinical studies. Overall, female mice have shown increased immunogenicity and are useful for assessing new molecular entities. However, this data indicates that all the lead vaccine adjuvants should be tested against male mice to rule out their suitability for clinical studies. 

Several adjuvants have been tested and approved in different vaccines. However, studies delineating sex differences are scarce. 

7DW8-5 is a well-known glycolipid vaccine adjuvant, providing its efficacy in several preclinical studies, particularly in vaccines for malaria.

Minor

Lines 34-36: The sentence should be rewritten. The present statement represents "the pathogen getting benefit from CD8 T cells". 

More references related to glycolipid mechanisms of action should be cited. 

Author Response

Lines 34-36: The sentence should be rewritten. The present statement represents "the pathogen getting benefit from CD8 T cells".  

As suggested, this sentence has been re-written. 

More references related to glycolipid mechanisms of action should be cited.   

As suggested, we added more references about glycolipid mechanisms of action.